# Optically Transparent Polydimethylsiloxane-Ethylene Oxide-Propylene Oxide Multiblock Copolymers Crosslinked with Isocyanurates as Organic Compound Sorbents

**DOI:** 10.3390/polym14132678

**Published:** 2022-06-30

**Authors:** Ilsiya M. Davletbaeva, Oleg O. Sazonov, Ilgiz M. Dzhabbarov, Ilnaz I. Zaripov, Ruslan S. Davletbaev, Alla V. Mikhailova

**Affiliations:** 1Technology of Synthetic Rubber Department, Kazan National Research Technological University, 68 Karl Marx St., Kazan 420015, Russia; sazonov.oleg2010@gmail.com (O.O.S.); ilgiz-9393@mail.ru (I.M.D.); zaripovilnaz@gmail.com (I.I.Z.); 2Department of Materials Science, Welding and Industrial Safety, Kazan National Research Technical University Named after A.N. Tupolev, Kazan 420111, Russia; darus@rambler.ru; 3Vernadsky Institute of Geochemistry and Analytical Chemistry of Russian Academy of Sciences, Moscow 119334, Russia; xemafiltra@yandex.ru

**Keywords:** multiblock copolymers, supramolecular structure, sorption activity, optical systems in analytical spectrometry, immobilization of organic reagents, test methods, water analysis, solid phase determination of elements

## Abstract

New crosslinked (polydimethylsiloxane-ethylene-propylene oxide)-polyisocyanurate multiblock copolymers (MBCs) were synthesized, and their supramolecular structure and sorption characteristics were studied. It was found that the interaction of PPEG and D_4_ leads to polyaddition of D_4_ initiated by potassium-alcoholate groups. The use of the amphiphilic silica derivatives associated in an oligomeric medium (ASiPs) leads to structuring of the MBC due to the transetherification reaction of the terminal silanol groups of the MBC with ASiPs. It was established that the supramolecular structure of an MBC is built according to the “core-shell” structure. The obtained polymers were tested as sorbents for the development of new methods for the concentration and determination of inorganic compounds. The efficiency of sorption of reagents increased with an increase in the “thickness” of the polydimethylsiloxane component of the “shell” and with a decrease in the size of the polyisocyanurate “core”. The use of the obtained polymers as adsorbents of organic reagents is promising for increasing the efficiency of field methods of chemical testing and inorganic analysis, including the determination of the elemental composition and the detection of traces of contamination.

## 1. Introduction

Block copolymers (BCs), including multiblock copolymers (MBCs) of amphiphilic nature, are currently attracting the efforts of researchers due to their ability to form various supramolecular structures. Block copolymers are a group of materials in which blocks from different polymer chains are chemically bonded. In most cases, the blocks are thermodynamically incompatible, but the covalent bonds between them lead to interactions, creating certain morphologies both in bulk and in solution. This behavior is a consequence of microphase separation caused by the thermodynamic incompatibility of the blocks [1,2,3,4,5,6,7,8,9,10,11].

The formation of self-assembled morphologies can be controlled by the composition of the block copolymer and the total chain length. For thin films of a block copolymer, additional interactions with a possible substrate and free interfaces should also be taken into account [12], whereas for assembly in solution, the formation of micelles of various shapes, lamellar structures, or the formation of voids also depend on the solubility parameters, possible crystallinity of the rod-forming block, the additional molecular components, etc. [13].

Block copolymers with precisely controlled molecular weight and narrow polydispersity, consisting of two or more chemically different and incompatible segments covalently bonded to each other, attract considerable attention due to their amazing ability to separate into many nanostructures in bulk and in solution [14,15,16,17,18]. For the simplest block copolymers, i.e., diblock copolymers, a variety of ordered structures are already known, such as classical body-centered cubic-packed spheres, hexagonally packed cylinders, alternating lamellae with a constant mean interface curvature, and complex morphologies such as bicontinuous gyroids and Frank–Kasper phases with non-constant mean curvatures [19]. By increasing the number of blocks, combinations of different elements can also be found, such as, for example, spheres and lamellae or cylinders and lamellae. Branched topologies such as stars have led to the formation of additional structures [20,21]. Wider controlled polydispersity of blocks can lead to stable morphologies, such as hexagonally perforated lamellae, which are unstable in narrowly dispersed diblock copolymers [22]. Such a combination of different segments of block copolymers has a huge potential for creating new superstructures and maintaining control over morphology [21,22,23].

In this series, polyorganosiloxane block copolymers should be discussed separately. The unique properties of polyorganosiloxanes [24] (hydrophobicity, gas permeability, low glass transition temperature, low surface energy) with hydrophilicity, thermal sensitivity, and bio- and hemocompatibility of polyoxyethylene oxide are manifested in siloxane-urethane-ethylene oxide block copolymers. As a rule, polyorganosiloxane block copolymers are heterogeneous systems. The degree of microphase separation can noticeably change as a result of varying the chemical nature of the blocks in the copolymer and their molecular weight, and has a significant effect on the diffusion properties of heterogeneous systems. Polyorganosiloxane block copolymers are synthesized by polymerization or polycondensation, depending on the origin of the starting reagents. The synthesis can include the stages of chemical assembly, including monomer–monomer, monomer–oligomer, monomer–monomer–oligomer, oligomer–oligomer, oligomer–monomer–oligomer, and oligomer–polymer [25]. Due to the availability of a wide range of reactive polysiloxanes and their excellent properties, the incorporation of polysiloxanes as “soft” segments in organic polymeric materials has been of great interest and has been used to prepare block, graft, or segmented copolymers for special applications.

Step polymerization is widely used for the synthesis of polysiloxane-containing segmented and multiblock copolymers. Hydroxyl and amine terminated polysiloxanes are versatile starting materials leading to silicone urethane, silicone urea, silicone ester, silicone amide, silicone imide, and other segmented/multiblock copolymers containing polysiloxane. 

Organosiloxane copolymers with a certain molecular weight can also be obtained by equilibrium copolymerization of organocyclosiloxanes in the presence of various molecular weight regulators [26]. Another possible method for the preparation of poly(organosiloxane) copolymers is the equilibrium copolymerization of organocyclosiloxanes with linear poly(organosiloxanes) [27,28,29,30]. In both cases, in addition to linear copolymers, the products contain significant amounts (more than 10%) of low molecular weight organocyclosiloxanes [31].

Block copolymers of polydimethylsiloxane and polyalkylene oxide are also of considerable interest for many technological solutions, from a component for polyurethane copolymers to surfactants in agrochemistry [32,33,34,35]. They have excellent surfactant properties and can reduce the surface tension of water to lower values than conventional hydrocarbon surfactants [36,37,38]. They have good thermal stability and, at the same time, they usually do not precipitate at low temperatures because the compounds cannot crystalize due to their highly branched structure. In addition, due to their molecular structure, the adhesion forces between individual molecules at phase boundaries are low, and for this reason the compounds are good wetting agents, and they are also excellent lubricants [39,40]. Silicone surfactants are known for their low toxicity in cosmetic applications because, due to their different structure, they cannot penetrate normal lipid bilayer structures [41].

Regarding amphiphilic organosiloxane block copolymers, they are widely covered in the literature [42,43]. Amphiphilic PDMS-POE diblock copolymers have been synthesized from hydrosilane-terminated PDMS and allyl-terminated POE homopolymers via an equimolar hydrosilylation reaction in the presence of a platinum catalyst. Similarly, polysulfone (PSF)-polyalkylene oxide-PDMS triblock copolymers were obtained by adding preformed α,ω-bis(hydrogensilyl)polydimethylsiloxane oligomers to allyl-terminated PSF-PAO [44,45]. Modification of biocompatible elastomers, especially polydimethylsiloxane (PDMS), by mixing with hydrophilic fillers used in various biomedical fields to increase the affinity of a hydrophobic polymer for water and increase the permeability of water vapor and bioactive substances was studied in [46]. 

One of the promising areas in which nanoporous materials, including those based on block copolymers, can be successfully used is transparent optical chemical sensors and test methods for the determination of various substances [47]. Such polymers can retain organic reagents, inorganic precipitants, enzymes, etc. In some cases, in the practice of analysis, sensors and sensitive elements on optically transparent polymer substrates can be more convenient and make it possible to visually observe the change in color and identify the content of certain metals. After the sorption of colored compounds or their formation on the surface, these polymers acquire a color characteristic of the compound being determined. Tests have been proposed for the detection of ions of such metals as, for example, zinc, lead, cobalt, copper, cadmium, mercury, nickel, and chromium. This approach is one of the promising directions in the development of applied research in order to obtain more sensitive sensors for the determination of various metals in the form of their colored complexes.

Silica gel is also widely used as substrates for chemical sensors. Silica gel is a porous, granular form of silica made synthetically from solutions of sodium silicate, silicon tetrachloride, or substituted chlorosilanes/orthosilicates. The active surface of silica gel with a large specific surface is important in adsorption and ion exchange. On the surface, the structure is either composed of siloxane groups (≡Si-O-Si≡) with an oxygen atom on the surface, or one of several forms of silanol groups (≡Si-OH). An optical chemical sensor of sulfur dioxide based on films of functional polymers for monitoring the air in the working area was considered in [48,49]. In [50,51], nanoporous materials were also considered as effective sorbents and substrates for analytical sensors.

Over the past three decades, the development of optical sensors continued. This is primarily due to their selectivity of the ions, which is due to the incorporation of highly selective ionophores into the membrane [52,53]. In addition to their selectivity and sensitivity, the production is simple and inexpensive, and these methods for determining metals proved themselves reliable in the field. Various copper optodes have been studied using fluorescence, absorption, and surface plasmon resonance. In a study, Ruiz et al. used a β-ketoimine calix(4)arene ionophore to detect the Cu(II) ion in aqueous solution using UV–visible spectrophotometry. They succeeded in developing a membrane selective towards Cu(II). However, the membrane must be stored under water to prevent it from drying out [54]. Another optical sensor for detecting Cu(II) based on fluorescence was developed by Aksuner et al. using a Schiff base as the ionophore. The sensor was found to be selective and sensitive to copper (II) ions with a faster response time than other metals, but in order to be stable over a longer period of time, it was stored in a THF desiccator in the dark. [55]. A more stable sensor was developed by Sandsem et al., immobilizing 4-decyloxy-2-(2-pyridylazo)-1-naphthol (DPAN) on a Nafion membrane (materials composed of perfluorosulfonic acid polymers). This modification of the original PAN-Nafion sensor is aimed at leaching the ionophore from the membrane. They incorporated this sensor into a controlled flow system that allowed them to analyze the concentration of copper in tap water. Another study [56] describes the selective determination of Cu(II) using a recently developed polymer membrane in a river water supply system through a continuous flow system.

A directed influence on the supramolecular organization of BC is a way to control both the physico-mechanical and physico-chemical properties of polymeric materials obtained on their basis. Promising for controlling both macromolecular and supramolecular structures are polymers based on 2,4-toluylene diisocyanate (TDI) and multiblock copolymers (MBCs) with polyorganosiloxane segments obtained by polyaddition of octamethylcyclotetrasiloxane (D_4_) to triblock copolymers of propylene and ethylene oxides (PPEG). 

In the work of [57], a high activity of the terminal potassium-alcoholate groups of PPEG was established in the reactions of the opening of TDI isocyanate groups, initiated by them according to the anionic mechanism. Depending on the reaction conditions created, the interaction of PPEG with TDI can be accompanied by the formation of both coplanar polyisocyanate blocks of acetal nature and branched polyisocyanurate structures. In [58], amphiphilic silica derivatives (ASiPs) (Figure 1) associated in an oligomeric medium were synthesized and characterized as modifiers of block copolymers based on macroinitiators and 2,4-toluene diisocyanate.

The aim of this work is to develop a route for the synthesis of optically transparent polydimethylsiloxane-ethylene oxide-propylene oxide block copolymers crosslinked with isocyanurates (IMBCs), which have a high sorption capacity.

To study the sorption capacity of the synthesized polymers, the La(III) complexes with arsenazo III were selected. The complex formation of the La(III) complexes with arsenazo III in solution has been well studied, which allows us to compare the obtained results of solid-phase sorption. However, the availability of reliable systems for the out-of-lab determination of the La(III) is limited.

## 2. Materials and Methods

### 2.1. Materials

The block copolymer of propylene and ethylene oxide (PPEG), with the formula HO[CH_2_CH_2_O]_n_[CH_2_(CH_3_)CH_2_O]_m_[CH_2_CH_2_O]_n_K where n ≈ 14 and m ≈ 51, had a molecular weight of 4200 g/mol and contained 30 wt% of peripheral polyoxyethylene blocks, where the content of potassium-alcoholate groups was 10.9% from the total number of functional groups, and was purchased from PJSC Nizhnekamskneftekhim (Nizhnekamsk, Russia). Octamethylcyclotetrasiloxane (D_4_) was purchased from Nanjing Union Silicon Chemical Co., Ltd. (USI Chemical, Nanjing, China). Amphiphilic branched silica derivatives associated with oligomeric medium (ASiPs) were obtained in the laboratory conditions [58]. Furthermore, 2,4-toluene diisocyanate ≥ 98 wt% (TDI) was purchased from Sigma-Aldrich (St. Louis, MO, USA). PPEG was additionally dried at a reduced pressure (approximately 0.1 kPa) and at an elevated temperature of 95 °C down to 0.01 wt% moisture concentration. Toluene was obtained from Component-reaktiv Ltd. (Moscow, Russia).

Rhodamine 6G (R6G), arsenazo III (AS III), conc. sulfuric acid, and LaCl_3_·7H_2_O were purchased from Sigma-Aldrich (St. Louis, MO, USA) and used as received. It is recommended to store the AS III solution in a dark vial and prepare a fresh solution every week. The overall absorbance of this dye solution slightly decreases over time; hence, the calibration curve measurements and the assay should preferably be conducted on the same day with the same AS III stock solution. LaCl_3_·7H_2_O was of ≥99.9% purity. Buffer, metal stock solutions, and dye stock solutions were made up in distilled water. Ethanol was purchased from MilliporeSigma (St. Louis, MO, USA).

### 2.2. Synthesis of MBCs and IMBCs

The reaction was carried out in two stages. In the first stage, multiblock copolymers (MBCs) were obtained through the interaction of PPEG with D_4_ in their molar relations, [D_4_]:[PPEG] = 0, 5, 10, 15. ASiPs were used as a modifier. The calculated amount of PPEG and D_4_, as well as ASiPs as a modifier, was placed in a round-bottom flask equipped with a reflux condenser and stirred for thirty minutes at T = 60 °C. Before conducting studies, MBCs were preheated under a residual pressure of 0.7 kPa until they reached a constant weight.

In the second stage, the IMBCs were obtained by the interaction of MBCs with TDI in various molar ratios, [PPEG]:[D_4_]:[TDI]. IMBCs were also prepared on the basis of PPEG and TDI in their molar ratios, [PPEG]:[TDI] = 1:8, 1:10. The reaction was carried out in toluene at 70 °C in a flask equipped with a reflux condenser. The polymerization process was carried out with constant stirring using a magnetic stirrer. The reaction mass was stirred at this temperature until complete dissolution of the PPEG or MBCs, and then introduced TDI. Five minutes after this, the polymer-forming system was poured into a Petri dish, and then it was cured at room temperature for 72 h.

### 2.3. Preparation of Samples

For the preparation of complexes of AS III with LaCl_3_ in an aqueous medium, the concentration of AS III was 1 × 10^−6^ mol/L. For immobilization, solutions of R6G and AS III were prepared by dissolving accurately weighed portions in ethyl alcohol with a concentration of R6G 1 × 10^−3^ mol/L and AS III with a concentration of 1 × 10^−4^ mol/L. R6G and AS III were immobilized on IMBCs by adsorption from their alcoholic solutions in a static mode with periodic stirring for 30 min. Then, IMBCs with R6G and AS III immobilized on them were kept in water for 120 min and dried. LaCl_3_ solutions with concentrations of 1 × 10^−1^, 1 × 10^−2^, 1 × 10^−3^, 1 × 10^−4^, and 1 × 10^−5^ g/dm^3^ were prepared by dissolving their exact weight in distilled water and then diluting them. The volume of the working solution was 40 mL. The polymer was kept in an aqueous solution of LaCl_3_ for 60 min. For aqueous solutions of LaCl_3_, the acidity values were pH = 7 and pH = 3. The pH of the medium was adjusted using sulfuric acid. The pH of the solutions was measured with a pH-673 potentiometer. We used the analytical electronic balance brand GH-252 (A&D Company ltd., Tokyo, Japan).

### 2.4. Measurements

#### 2.4.1. NMR Spectroscopy

The ^1^H NMR spectra were measured on the Bruker AC400 (400 MHz) spectrometer (Billerica, MA, USA) at 20 °C, using deuterated chloroform solutions.

#### 2.4.2. Fourier-Transform Infrared (FTIR) Spectroscopy Analysis

The FTIR spectra of the products were recorded on an InfraLUM FT-08 Fourier-transform spectrometer (Lumex, St. Petersburg, Russia) using the attenuated total reflection technique. The spectral resolution was 2 cm^−1^, and the number of scans was 16.

#### 2.4.3. Thermal Gravimetric Analysis (TGA)

TGA was performed using the STA-6000 TG-DTA combined thermal analyzer (PerkinElmer, Waltham, MA, USA). The samples (0.1 g) were loaded in alumina pans and heated from 30 °C to 750 °C at a rate of 5 K/min in a nitrogen atmosphere.

#### 2.4.4. Tensile Stress–Strain Measurements

Tensile stress–strain measurements were obtained from the film samples of size 40 mm × 15 mm with the Universal Testing Machine Inspekt Mini (Hegewald & Peschke Meß- und Prüftechnik GmbH, Nossen, Germany) at 293 ± 2 K, 1 kN. The crosshead speed was set at 50 mm·min^−1^ and the test continued until sample failure. Minima of five tests were analyzed for each sample and the average values were reported based on ASTM D 882.

#### 2.4.5. AFM Studies

Surface topography of samples was imaged on an atomic force microscope (AFM) Nano-DST (Pacific Nanotechnology, Santa Clara, CA, USA), operated in semi-contact (tapping) mode under ambient conditions. NSG01 probes (TipsNano, Tallinn, Estonia) were used with the following parameters: n-Si, tip curvature 10 nm, frequency ca. 150 kHz, and force constant 5.1 N/m.

#### 2.4.6. Ultraviolet Visible (UV–Vis) Spectroscopy

Ultraviolet visible (UV–Vis) spectroscopy was performed with the Double-beam Spectrophotometer Specord 210 plus (Analytik Jena, Germany) in quartz cells with a length of 10 mm, scan step of 1 nm, and scan speed of 10 nm/s. Calculation of ΔAbs. Was carried out using the equation:$$\Delta Abs.=Abs._{1}-Abs._{0}$$
where *Abs.*_1_ is the absorption coefficient of the AS III complex with La(III), and *Abs.*_0_ is the absorption coefficient of AS III.

### 2.5. Water Absorption Tests

Water uptake tests in accordance with ISO 20344/2011 were performed in order to investigate the effect of water absorption on the properties of NR rubber composites reinforced with natural fiber. Water absorption (uptake) experiments were conducted on circular samples with a 30 mm diameter and 2 mm thickness. At least five tests were realized for every group/type of samples. Before initial weighing, all samples were dried for 24 h in a laboratory oven at 80 °C and then placed in a desiccator for cooling. Water absorption tests were conducted by immersing the samples in distilled water in closed glass containers and keeping them at a room temperature of 23 ± 2 °C. The samples were taken out from the glass containers at regular time intervals, and the wet surfaces were quickly wiped using a tissue paper and weighed until no further increase in water absorption was detected. The weighing was performed in a bottle with a stopper and the weighing process of every sample was no longer than 10 s. The weighing precision was within 0.1 mg.

The water absorption in the polymer samples at the time *t*, *Q_t_,* expressed in wt%, was calculated by finding the difference between the weights of samples immersed in water (*W_t_)* and dry samples (*W*_0_), using the following equation:$$Q_{t}=\frac{\left(W_{t}-W_{0}\right)}{W_{0}}\times 100\%$$

## 3. Results

### 3.1. Polymer Characterization

The sequence of synthesis was the preliminary preparation of a multiblock copolymer (MBC) by copolymerization of PPEG with D_4_ initiated by terminal potassium-alcoholate groups. Then, TDI was added into the reaction system. ASiPs were used to impact to the supramolecular organization of the obtained polymers [58,59].

Since polydimethylsiloxane (PDMS) and PEG are thermodynamically incompatible polymers, the copolymerization of PPEG with D_4_ initiated by terminal potassium-alcoholate groups was identified by homogenization of the reaction system. In the case of homopolymerization of D_4_, the phase interface should be traced. In the investigated reaction system during the interaction of PPEG with D_4_, the viscosity increased and a homogeneous opaque mass formed. In order to establish the time of completion of the reaction, a test, which consisted of comparing the mass of the resulting polymer with the mass of the initial reaction system, was carried out. For this purpose, the polymer-forming system was under vacuum pressure at P = 0.07 kPa and T = 90–100 °C for one hour to distill unreacted D_4_ before the weighing. The reaction was stopped when the monomer conversion was constant.

The formation of polydimethylsiloxane segments in the MBCs obtained at [D_4_]:[PPEG] = 5, 10, and 15 is confirmed by ^1^H NMR spectra (Figure 2).

The number of octamethyltetrasiloxane units (*n*) in the MBC composition was calculated using the ratio of integral intensities of proton signals related to methyl groups of the propylene oxide block of PPEG (d, 1.14 ppm) and methyl groups of the polydimethylsiloxane block (e, 0.08 ppm) in the studied multiblock copolymer samples. The calculations were carried out using information on the content of the polypropylene oxide block (70%) and the molecular weight (4200 Da) of the commercially available PPEG. Conversion of D_4_ was determined as a ratio of calculated and experimental ratios of D_4_ and PPEG (Table 1).

From the ratio of the proton signals corresponding to the methyl group at the silicon atom (e) and the methyl group belonging to the polyoxypropylene units (d), the number of octamethyltetrasiloxane units (*n*) in the MBC composition and the conversion of D_4_ as a ratio of the calculated and experimental ratios of D_4_ and PPEG were calculated (Table 1). According to the calculations, with an increase in the molar excess of D_4_ relative to PPEG, the D_4_ conversion increases (Table 1). A comparison of the octamethyltetrasiloxane units (*n*) and the molar ratio [D_4_]:[PPEG] at which these values were achieved allows for the conclusion that almost all PEG macromolecules participate in the polyaddition of D_4_.

Interaction of MBCs with TDI is accompanied by complete consumption of isocyanate groups. This is evidenced by the absence of analytical bands in the region of 2275 cm^−1^ on the RTIR spectra. The formation of polyisocyanurates is evidenced by intensive analytical bands in the region of 1700 and 1410 cm^−1^, corresponding to the C=O bond of isocyanurates. The formation of a small number of urethane groups was identified by the presence of a low-intensity shoulder in the 1730 cm^−1^ region.

Thus, as a result of the polyaddition of D_4_ initiated by terminal potassium-alcoholate groups to PPEG, MBC formation occurs (Figure 3). The interaction of MBCs with 2,4-toluene diisocyanate leads to the subsequent formation of a branched structure of IMBCs. The migration of potassium ions in the zone of isocyanate groups of TDI interaction creates active centers that cause the formation of isocyanurates. As a result of the sequence of chemical reactions, isocyanurate cycles, the initiated formation of which occurs at the active centers of MBCs, are combined into a single polyisocyanurate framework, creating a “core” along the periphery of which a shell consisting of MBCs is laid (Figure 4). Potassium ions in the composition of IMBCs are captured in the cavity created by the open-chain analogue of crown ethers. As part of the PPEG, polyoxyethylene segments can perform this function.

Polyisocyanurates have relatively high heat resistance. The temperature of their thermal degradation lies in the region of 350 °C. The TGA curves show no mass loss up to the temperature region of 230 °C. After achieving 5 wt% weight loss, the next thermal degradation region begins at T = 340 °C. The use of ASiPs has practically no effect on the thermo-resistance of IMBCs. The results of thermogravimetric analysis indicate a significant contribution of polyisocyanurate fragments to the high heat resistance of the studied polymers.

With a relatively high molar excess of TDI ([PPEG]:[D_4_]:[TDI] = 1:X:10), the density of the nodes of the IMBC polymer framework due to the polyisocyanurates formed here is so high that an increase in the molar excess of D_4_ does not lead to a change in the stress–strain tests (Figure 5a).

In order to describe the mechanical properties of obtained polymers and the influence of D_4_ and TDI content on these properties, stress–strain measurements were performed. Thus, the significant impact of these ratios on mechanical properties, as well as on the supramolecular structure of obtained polymers, was identified based on results of mechanical property tests and additional direct and indirect methods described earlier.

According to the obtained results, synthesized polymers show good elastic properties and mechanical strength that are very important for modern polymer materials. Thus, the elasticity of polymers obtained in all ranges of TDI ratios show values close to 100% with simultaneous mechanical strength higher than 4 MPa.

With a decrease in the molar excess of TDI from the ratio [PPEG]:[D_4_]:[TDI] = 1:X:10 (Figure 5a) to the ratio [PPEG]:[D_4_]:[TDI] = 1:X:8 (Figure 5b), the strength of the IMBC based on [PPEG]:[D_4_]:[TDI] = 1:X:8 drops almost 2 times compared to the IMBC obtained with [PPEG]:[D_4_]:[TDI] = 1:X:10. Such a sharp drop in polymer strength with a decrease in TDI content can be explained by the fact that at the molar ratio [PPEG]:[D_4_]:[TDI] = 1:X:8, the most favorable conditions are created for combining their own microphase of flexible chain blocks outside the zone of isocyanurate nucleus formation. Because the resulting polydimethylsiloxane component does not have direct binding to the polyisocyanurate rigid “core” and is located along the periphery of the supramolecular structure built according to the “core-shell” type, it stretches the PPEG macrochain at an increasing distance from the “core.” As a result, the first layer of the “shell” consists of polyoxyethylene segments directly associated with the “core.” Next is a layer of associated polyoxypropylene segments. Terminal polyoxyethylene segments directly associated with polydimethylsiloxane blocks are also isolated into their own microphase due to their thermodynamic incompatibility with the PDMS component of the IMBC chain.

As the content of ASiPs increases during IMBC synthesis, there is a noticeable decrease in the strength of samples obtained at the molar ratio [PPEG]:[D_4_]:[TDI] = 1:15:8 (Figure 6a). This may be due to the drawing of peripheral polydimethylsiloxane chains from the “shell” into the interglobular space as a result of the interetherification reaction (Figure 7).

In contrast, for polymers obtained at the molar ratio [PPEG]:[D_4_]:[TDI] = 1:15:10, the introduction of the ASiP modifier does not lead to a decrease in the strength of the samples (Figure 6b). However, due to the elongation of the polydimethylsiloxane block as a result of transetherification reactions involving ASiPs, an increase in the elongation values of the samples to the point of mechanical destruction is observed.

### 3.2. Surface Morphology of IMBC

To describe the surface morphology of the obtained polymers and to identify voids and globules on the outer surface, measurements were made using atomic force microscopy (AFM). According to Figure 8a, for the IMBC obtained at the molar ratio [PPEG]:[D_4_]:[TDI] = 1:15:8, voids are observed in the AFM image against the background of morphology that could be identified as globular.

For samples obtained at the molar ratio [PPEG]:[D_4_]:[TDI] = 1:15:8 and using [ASiP] = 0.2 wt%, void disappearance is observed, but globular morphology of the IMBC surface is preserved (Figure 8b). When the ASiP content is increased to 1.0 wt%, a further change in the surface morphology of the IMBC obtained at the molar ratio [PPEG]:[D_4_]:[TDI] = 1:15:8 is observed (Figure 8c).

Based on the results of surface morphology and their comparison with the results of mechanical measurements, NMR and sorption capacity, the hypothesis about the supramolecular structure built according to the “core-shell” type was confirmed. 

### 3.3. Polymer Sorption Capacity

The presence of voids due to the globular supramolecular structure and the specific “loose” packaging of polydimethysiloxane macrochains located on the outer surface of the “core-shell” structure creates a prospect for the study of the obtained polymers as effective sorbents for organic reagents.

Patterns of change in free volume were studied by measuring the maximum degree of water absorption of the IMBCs (Table 2, Table 3 and Table 4). It was found that the water absorption for the studied samples is not accompanied by a change in their geometric dimensions. That is, samples in water do not swell, but absorb water due to the existence of voids. According to the results obtained, a decrease in the length of the PDMS segment, an increase in the content of polyisocyanurates, and an increase in the content of the ASiP modifier leads to a decrease in water absorption values.

Rhodamine 6G was used as a dye to study the sorption properties of polymers. Figure 9a shows a significant increase in the amount of dye immobilized on IMBC film samples as the mole ratio of D_4_ increases. The results confirm the fact that PDMS segments that are part of the MBC branches make a significant contribution to the supramolecular organization of the IMBC.

The fact that a significant contribution to the formation of the supramolecular structure of the resulting polymers is made by their modification using ASiPs can be identified by the electronic absorption spectra shown in Figure 9b. Thus, as the ASiP content increases, there is a marked decrease in the absorption intensity in the maximum region of the electron absorption spectrum (540 nm) due to the R6G immobilized on the polymers. The decrease in absorption intensity with an increase in ASiP content suggests that the interglobular space in the polymer is filled with the polydimethylsiloxane component. As a result of the decrease in free volume in the polymer, their sorption capacity also decreases.

### 3.4. Study of IMBC as a Matrix for Immobilization of Analytical Organic Reagents Used in Expression and Test Methods

Organic analytical reagents (ORs) are able to react with detectable metal ions and form colored complexes. Currently, similar complexation reactions with immobilized ORs on other matrices are quite well studied [60,61,62]. Complex formation reactions are accompanied by a change in color of the reaction systems and, at optimal pH, are the basis for selective determination of metal cations both in solutions and on solid matrices. Polymer matrices for sorption of complex organic compounds are known and widely used. However, polymeric substrates do not always have a porous structure. As a result, their low sorption capacity and low efficiency are observed when used for analytical purposes.

In this paper, arsenazo III (AS III) (Figure 10) was used as an OR, and water-soluble lanthanum salts (LaCl_3_) were used as analytes. In an aqueous solution at pH = 3÷3.8, the maximum wavelength of the complex La(III) with arsenazo III was 650 nm.

For the purpose of comparative analysis, electronic spectra were obtained (Figure 11a) and calibration curves for aqueous solutions of AS III and complexes of La(III) with AS III were constructed (Figure 11b).

As a polymer matrix for sorption of AS III and subsequent qualitative and quantitative determination of La(III) ions, the IMBC obtained at [PPEG]:[TDI] = 1:8, [PPEG]:[D_4_]:[TDI] = 1:15:8 was used. Similar IMBCs obtained in the presence of 0.2 wt% ASiP were also used. AS III was firstly sorbed on an IMBC from an alcohol solution. Then, film samples were sorbed from an aqueous solution of La(III). Absorption spectra of complexes La(III) with AS III formed on a polymer substrate are shown in Figure 12, Figure 13, Figure 14 and Figure 15. 

According to the analysis of absorption spectra and calibration plots, sorption by polymers at pH = 3 is more efficient. At pH = 7, where AS III is in ionic form, competition of functional groups of the organic analytical reagent between the solid phase and metal ions can occur. It seems that at pH = 7, the functional groups of the reagent are more closely related to the polymer.

According to the results, the IMBC of the composition [PPEG]:[D_4_]:[TDI] = 1:15:8 in the presence of [ASiP] = 0.2 wt% showed the highest sorption capacity compared to other test samples (Figure 15). The polymer based on [PPEG]:[TDI] = 1:8 (without D_4_) showed significantly lower sorption capacity compared to IBMCs. The calibration curves were linear for the entire study range of La(III) ion concentrations.

Thus, an advantage of using IMBCs as a substrate for analysis is the ability to determine, even in a neutral environment, very low, practically trace amounts of La(III) ions.

## 4. Conclusions

Multiblock copolymers prepared using PPEG, D_4_, and TDI were investigated. It was shown that during the interaction of PPEG and D_4_, the polyaddition of D_4_ initiated by potassium-alcoholate groups occurs. It was found that an increase in the relative content of D_4_ leads to an increase in the size of the polydimethylsiloxane block in the MBC composition. The use of ASiPs leads to the structuring of MBCs by a transetherification reaction involving terminal silanol groups of MBCs and ASiPs. The molar ratio of PPEG, D_4_, and TDI was established, in which the supramolecular structure of the IMBC is formed by the “core-shell” type. Modification of IMBCs using ASiPs has a marked effect on their supramolecular structure. As a result of the transetherification reaction involving terminal silanol groups of MBCs and ASiPs, the polydimethylsiloxane component fills the free space between globular formations.

The features of the supramolecular organization of IMBCs affect their sorption capacity. It was found that the sorption efficiency of organic reagents increases with an increase in the content of polydimethylsiloxane segments in the composition of the studied polymers and a decrease in the proportion of polyisocyanurates. IMBCs were investigated as a template for immobilization of the analytical organic reagent AS III used in the express and test methods. The linearity range of calibration curves when determining La(III) ions with immobilized AS III on IMBCs of any proposed composition is 5 orders of magnitude of the concentration determined in g/dm^3^. The color reaction on the IMBC polymer sorbent is reproducible, the formed polymer complex is stable for a long time, and test systems are easily applicable when working in the field. The observed higher efficiency of determination of La(III) ions with immobilized AS III is due to the fact that concentration of metal ions occurs in the IMBC matrix.

## Figures and Tables

**Figure 1 polymers-14-02678-f001:**
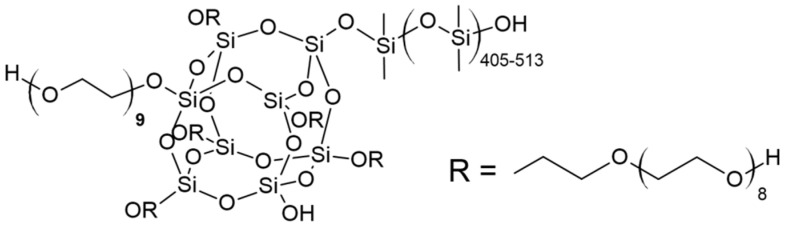
The structure of ASiP.

**Figure 2 polymers-14-02678-f002:**
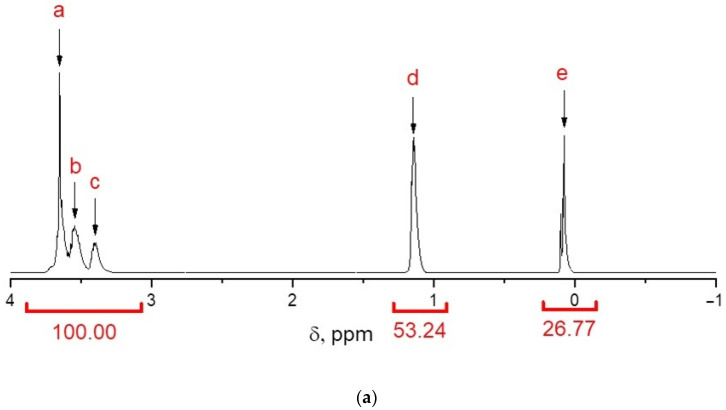
^1^H NMR spectra for MBCs obtained at molar ratios [D_4_]:[PPEG] = 5 (**a**), 10 (**b**), 15 (**c**).

**Figure 3 polymers-14-02678-f003:**
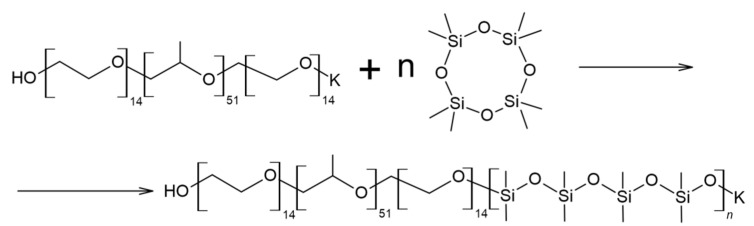
Scheme of MBC synthesis.

**Figure 4 polymers-14-02678-f004:**
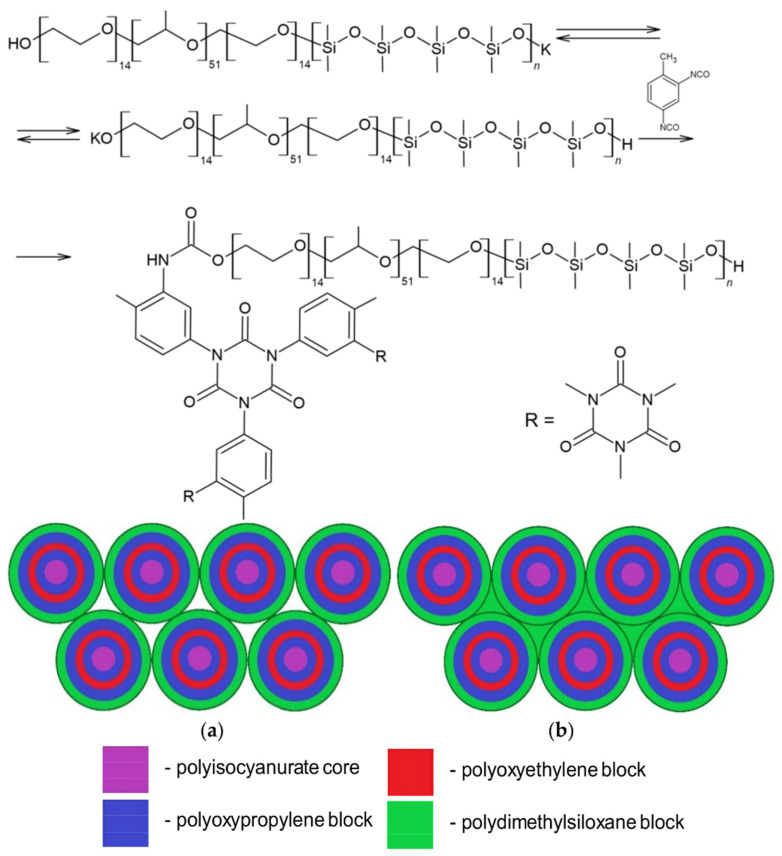
Scheme of IMBC synthesis and supramolecular “core-shell” structure (**a**,**b**) formation.

**Figure 5 polymers-14-02678-f005:**
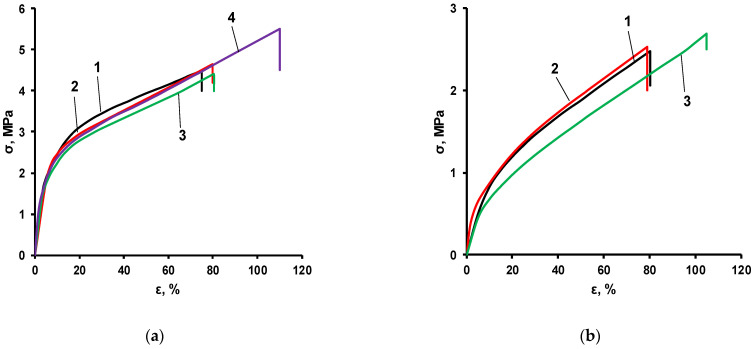
Stress–strain tests for IMBCs obtained at [PPEG]:[D4]:[TDI] = 1:0:10 (1), 1:5:10 (2), 1:10:10 (3), 1:15:10 (4) (**a**); [PPEG]:[D4]:[TDI] = 1:0:8 (1), 1:2:8 (2), 1:15:8 (3) (**b**).

**Figure 6 polymers-14-02678-f006:**
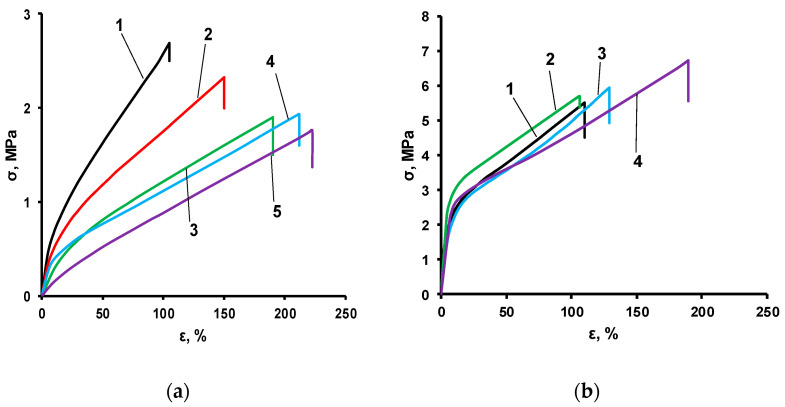
Stress–strain tests for IMBCs obtained at [PPEG]:[D4]:[TDI] = 1:15:8 [ASiP] = 0 (1), 0.2 (2), 0.4 (3), 0.7 (4), 1.0 (5) wt% (**a**) and [PPEG]:[D4]:[TDI] = 1:15:10 [ASiP] = 0 (1), 0.2 (2), 0.4 (3), 0.7 (4), 1.0 (5) wt% (**b**);

**Figure 7 polymers-14-02678-f007:**
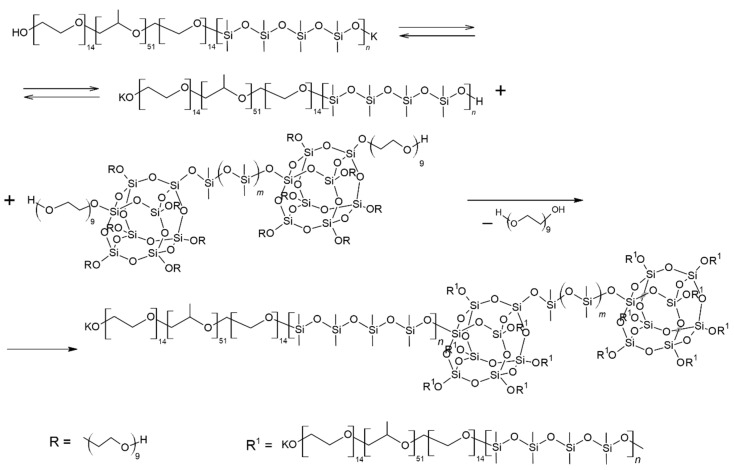
Scheme of interetherification reaction involving MBCs and ASiPs.

**Figure 8 polymers-14-02678-f008:**
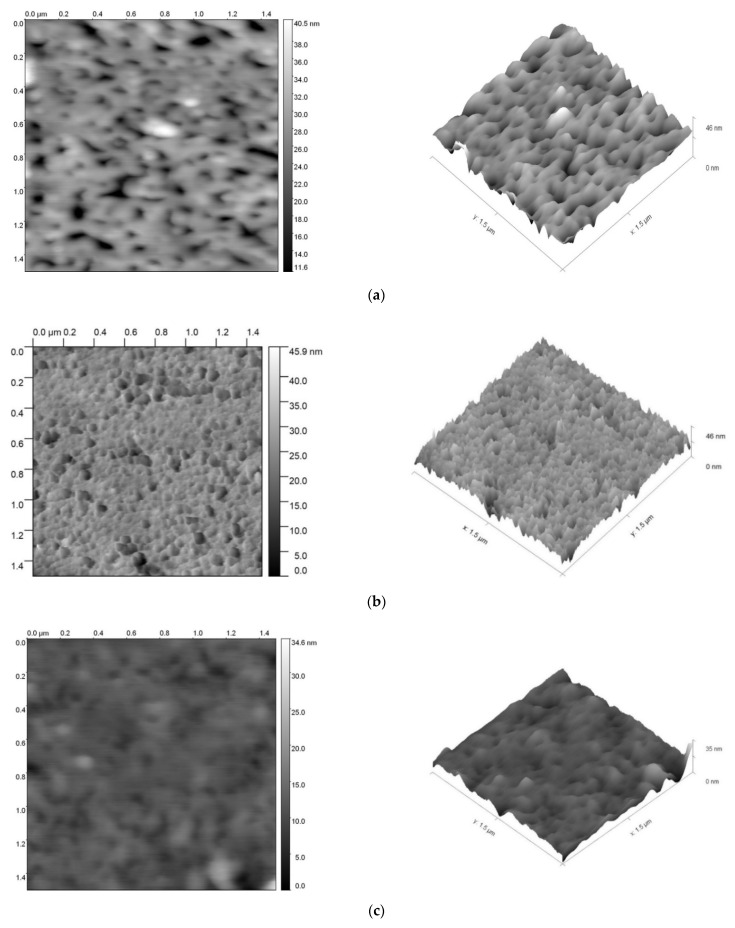
AFM images for IMBCs obtained at [PPEG]:[D_4_]:[TDI] = 1:15:8 (**a**) and [ASiP] = 0.2 (**b**), (**c**) 1.0 wt%.

**Figure 9 polymers-14-02678-f009:**
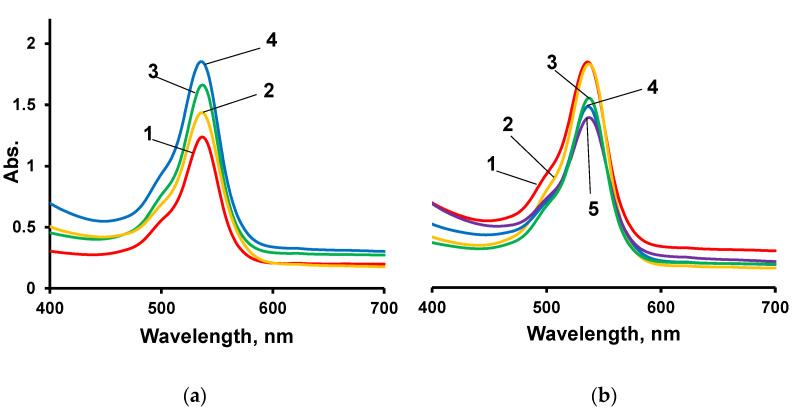
Electronic spectra of R6G immobilized on IMBCs obtained at [PPEG]:[D_4_]:[TDI] = 1:0:8 (1), 1:5:8 (2), 1:10:8 (3), 1:15:8 (4) (**a**) and [PPEG]:[D_4_]:[TDI] = 1:15:8, [ASiP] = 0 (1), 0.2 (2), 0.4 (3), 0.7 (4), 1.0 (5) wt% (**b**).

**Figure 10 polymers-14-02678-f010:**
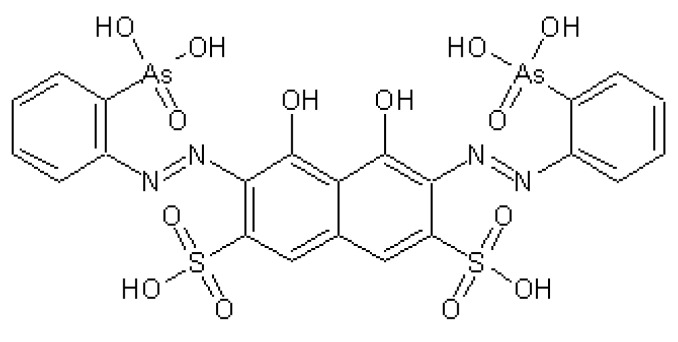
Chemical structure of arsenazo III.

**Figure 11 polymers-14-02678-f011:**
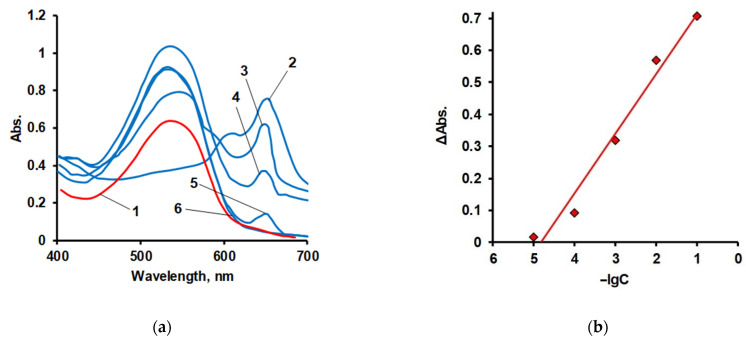
Electronic spectra of AS III aqueous solution (1), aqueous solutions of complexes of AS III with La(III), [LaCl_3_] = 1 × 10^−1^ (2), 1 × 10^−2^ (3), 1 × 10^−3^ (4), 1 × 10^−4^ (5), 1 × 10^−5^ (6) g/dm^3^ (**a**) and calibration curves for complexes of AS III with La(III) at λ = 650 nm (**b**) pH = 7.

**Figure 12 polymers-14-02678-f012:**
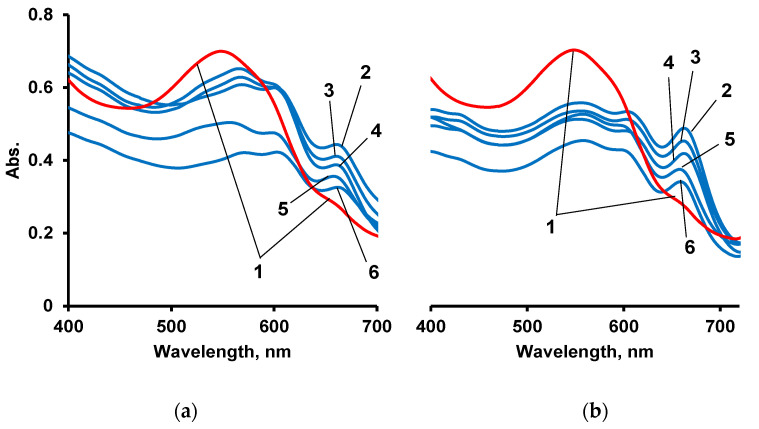
Electronic spectra of AS III (1) and complexes of AS III with La(III) sorbed on IMBC obtained at [PPEG]:[D_4_]:[TDI] = 1:15:8. [LaCl_3_] = 1 × 10^−1^ (2), 1 × 10^−2^ (3), 1 × 10^−3^ (4), 1 × 10^−4^ (5), 1 × 10^−5^ (6) g/dm^3^ pH = 7 (**a**) and [PPEG]:[D_4_]:[TDI] = 1:15:8. [LaCl_3_] = 1 × 10^−1^ (2), 1 × 10^−2^ (3), 1 × 10^−3^ (4), 1 × 10^−4^ (5), 1 × 10^−5^ (6) g/dm^3^ pH = 3 (**b**).

**Figure 13 polymers-14-02678-f013:**
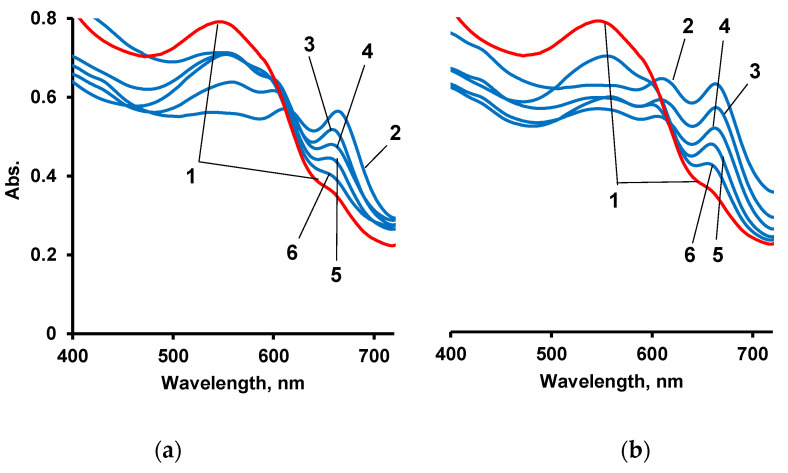
Electronic spectra of AS III (1) and complexes of AS III with La(III) sorbed on IMBC obtained at [PPEG]:[D4]:[TDI] = 1:15:8 with 0.2 wt% ASiP. [LaCl3] = 1 × 10^−1^ (2), 1 × 10^−2^ (3), 1 × 10^−3^ (4), 1 × 10^−4^ (5), 1 × 10^−5^ (6) g/dm^3^ pH = 7 (**a**) and [PPEG]:[D4]:[TDI] = 1:15:8 with 0.2 wt% ASiP. [LaCl_3_] = 1 × 10^−1^ (2), 1 × 10^−2^ (3), 1 × 10^−3^ (4), 1 × 10^−4^ (5), 1 × 10^−5^ (6) g/dm^3^ pH = 3 (**b**).

**Figure 14 polymers-14-02678-f014:**
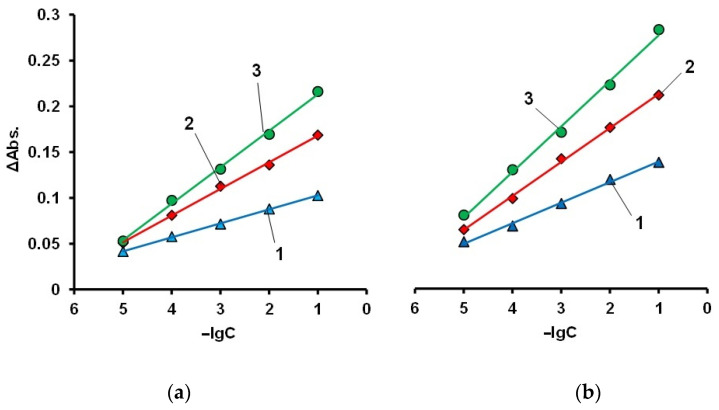
Calibration curves at λ = 650 nm for AS III complexes with La(III) sorbed on IMBCs obtained at [PPEG]:[TDI] = 1:8 (1), [PPEG]:[D_4_]:[TDI] = 1:15:8 (2), [PPEG]:[D_4_]:[TDI] = 1:15:8 with 0.2 wt% ASiP (3) pH = 7 (**a**) and [PPEG]:[TDI] = 1:8 (1), [PPEG]:[D_4_]:[TDI] = 1:15:8 (2), [PPEG]:[D_4_]:[TDI] = 1:15:8 with 0.2 wt% ASiP (3) pH = 3 (**b**).

**Figure 15 polymers-14-02678-f015:**
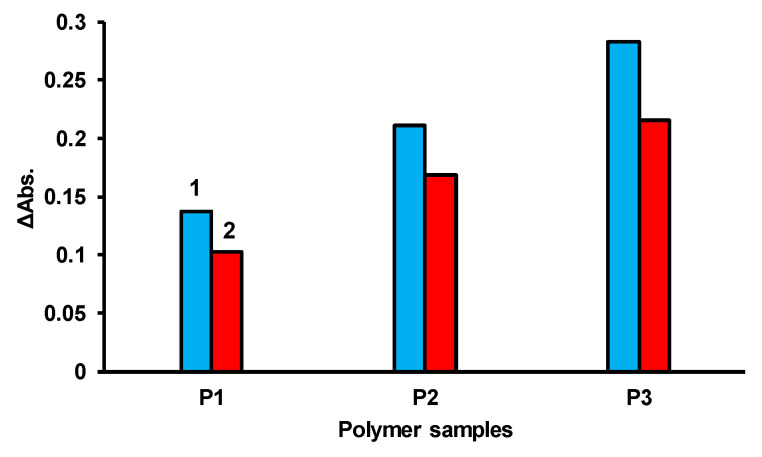
Sorption efficiency of AS III complex with La(III) on IMBCs obtained at [PPEG]:[D_4_] = 1:8 (P1), [PPEG]:[D_4_]:[TDI] = 1:15:8 (P2), [PPEG]:[D_4_]:[TDI] = 1:15:8 with [ASiP] = 0.2 wt% (P3), [LaCl_3_] = 1 × 10^−1^ g/dm^3^, pH = 3 (1), pH = 7 (2).

**Table 1 polymers-14-02678-t001:** Calculated number of octamethyltetrasiloxane units (*n*) in MBC and D4 conversion.

[D_4_]:[PPEG]	*n*	D_4_ Conversion, %
5	3.1 (3)	62
10	7.6 (8)	76
15	14.4 (14)	96

**Table 2 polymers-14-02678-t002:** Water absorption (wt%) for IMBC obtained at [PPEG]:[D_4_]:[TDI] = 1:15:X.

Reagent Content	[TDI] (X)				
	8	9	10	12	15
Water absorption, wt%	26	23	21	17	14

**Table 3 polymers-14-02678-t003:** Water absorption (wt%) for IMBC obtained at [PPEG]:[D_4_]:[TDI] = 1:X:8 and 1:X:10.

Reagent Content		[TDI]	
		8	10
[D_4_] (X)	0	21	17
	5	23	20
	10	24	21
	15	26	22

**Table 4 polymers-14-02678-t004:** Water absorption (wt%) for IMBC obtained at [PPEG]:[D_4_]:[TDI] = 1:15:8 и 1:15:10 in dependence of ASiP content.

Polymer Sample	[ASiP], wt%			
	0.2	0.4	0.7	1
[PPEG]:[D_4_]:[TDI] = 1:15:8	26	25	23	22
[PPEG]:[D_4_]:[TDI] = 1:15:10	22	21	20	10

## Data Availability

The data presented in this study are available on request from the corresponding author.

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
