# Peer review of "Optically Transparent Polydimethylsiloxane-Ethylene Oxide-Propylene Oxide Multiblock Copolymers Crosslinked with Isocyanurates as Organic Compound Sorbents"

_polymers, 2022, doi:10.3390/polym14132678_

Round 1

Reviewer 1 Report

The work entitled “Optically transparent polydimethylsiloxane-ethylenoxide-propylene oxide multiblock copolymers crosslinked with isocyanurates as organic compounds sorbents” reports on the synthesis of new crosslinked multiblock copolymers and evaluates their supramolecular structure and sorption characteristics. The subject is well introduced. However, it is way too long. The introduction should be significantly reduced so not to lose the attention of the readers. In fact, this is the only problematics with this manuscript. The methodology is extremely well prepared, very detailed and clear enough so others can replicate their work. The discussion is focus, very easy to understand and leaves no questions to answer regarding the collected data. Most importantly, the authors supported their discussion with the literature, improving the quality of the overall work.

Author Response

Dear Reviewer,

Thank you for your comments and feedback, which have helped us to substantially improve our manuscript. We carefully investigated you recommendations and prepared comments for each item you mentioned in your review.

  1. The introduction should be significantly reduced so not to lose the attention of the readers.

Answer

The introduction part rethought and significantly changed in order to improve the general perception of the manuscript.

Reviewer 2 Report

Block copolymers has good research value and significance due to the ability to form various supramolecular structures. Authors synthesized a new crosslinked (polydime thylsiloxane-ethylene-propylene oxide)-polyisocyanurate multiblock copolymers, and studied the supramolecular structure and sorption characteristics. Some results have been carried out. However, the current form of this study cannot be acceptable. Some aspects as listed below:

1. In Page 7 line 295, what is the Wt and W0? It should be given.

2More details about the tensile stress – strain measurements should be given. Why 50 mm/min is used?

3. In figure 5, 6, the image quality should be improved.

4. What is the purpose of the surface morphology?

5. More discussion should be given.

Author Response

Dear Reviewer,

Thank you for your comments and feedback, which have helped us to substantially improve our manuscript. We carefully investigated you recommendations and prepared comments for each item you mentioned in your review:

  1. In Page 7 line 295, what is the Wt and W0? It should be given.

Answer:

Description added in the manuscript

  1. More details about the tensile stress – strain measurements should be given. Why 50 mm/min is used?

Answer:

The current loading speed was defined based on experiments, experience and shows better correspondence between elastic deformation and destruction. This indirect method with combination with another direct one as NMR and AFM allow us to describe the molecular and supramolecular structure of obtained molecules. Taken into consideration the results obtained based on variety of methods we conclude that a significant drop in polymer strength with a decrease in TDI content can be explained by the fact that at the molar ratio [PPEG]:[D4]:[TDI] = 1:X:8, the most favorable conditions are created for combining microphase of flexible chain blocks outside the zone of isocyanurate nucleus formation.

Additional explanations was added to the manuscript.

  1. In figure 5, 6, the image quality should be improved.

Answer:

The images quality improved.

  1. What is the purpose of the surface morphology?

Answer:

AFM measurements were used to describe the surface morphology of obtained polymers and identify voids and globules on the outside surface. Based on results of surface morphology and collating them with results of mechanical measurements, NMR and sorption capacity, the hypothesis regarding structure of polymers prepared.

  1. More discussion should be given.

Answer:

Additional explanation added to the Result and discussion part.

Round 2

Reviewer 2 Report

Block copolymers has good research value and significance due to the ability to form various supramolecular structures. Authors have revised the manuscript according to the comments. However, the current form of this study cannot be acceptable. Some aspects as listed below:

 1. In Table 1, more details about the calculation should be given in the paper.

 2The tensile standard is suggested to be given.

 3. More discussion about the mechanical performance should be given.

Author Response

Dear Reviewer,

Thank you for your comments and feedback, which have helped us to substantially improve our manuscript. We carefully investigated you recommendations and prepared comments for each item you mentioned in your review.

  1. In Table 1, more details about the calculation should be given in the paper.

Answer.

More detailed information and description of calculation added to the manuscript

 2.  The tensile standard is suggested to be given

Answer

Standard for mechanical properties measurement added

  1. More discussion about the mechanical performance should be given.

Answer

Additional information regarding mechanical properties added to the text